# Mie Resonance Engineering in Two Disks

Evgeny Bulgakov [1,2,†], Konstantin Pichugin [1,†] and Almas Sadreev [1,*,†]

1   L.V. Kirensky Institute of Physics, Federal Research Center KSC Siberian Branch, RAN,
    660036 Krasnoyarsk, Russia; ben@tnp.krasn.ru (E.B.); knp@tnp.krasn.ru (K.P.)
2   Reshetnev Siberian State University of Science and Technology, 660037 Krasnoyarsk, Russia
*   Correspondence: almas@tnp.krasn.ru
†   These authors contributed equally to this work.

**Abstract:** Recently the recipes to achieve the high-Q subwavelength resonances in an isolated dielectric disk have been reported based on avoided crossing (anticrossing) of the TE resonances under variation of the aspect ratio of the disk. In a silicon disk that recipe gives an enhancement of the $Q$ factor by one order of magnitude. In the present paper we present the approach based on engineering of the spherical Mie resonances with high orbital index in two coaxial disks by two-fold avoided crossing of the resonant modes of the disks. At the first step we select the resonant modes of single disk which are degenerate because of the opposite symmetry. Approaching of the second disk removes this degeneracy because of interaction between the disks. As a result at certain distances we realize the hybridized anti-bonding resonant modes whose morphology becomes close to the spherical Mie resonant mode with high orbital index. Respectively the $Q$ factor of the anti-bonding resonant mode can be enhanced by three orders of magnitude compared to the case of single disk.

**Keywords:** resonant modes; avoided crossing; mie modes

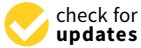



## 1. Introduction

In 1908 Gustav Mie derived the analytical solution of Maxwell equation in a presence of spherical particles and analyzed the scattering problem of wavelength-comparable particles under the excitation of electromagnetic (EM) plane wave, which afterward was well known as the Mie solution [1]. A century later, the current research of ubiquitous scattering phenomena in artificially engineered micro/nanostructures has been actively developing from the start point of Mie scattering. Each Mie solution is specified by $m$-th order electric and magnetic multipolar modes. The imaginary part of complex Mie eigenfrequency is responsible for radiation losses while the ratio of real part and imaginary part defines the $Q$ factor of the mode. Because of the minimal area of the sphere's surface at the given volume any compact dielectric resonator of the same volume with its shape different from the sphere has the lower $Q$ factor. Nevertheless, disk or cuboid are of interest from the standpoint of implementation into optical circuits and fabrication convenience. However, no intuitive fundamental understanding of the optical resonance in non-spherical structures is available, which has substantially delayed the device development with dielectric materials.

Recently a considerable progress in understanding of enhancement of the $Q$ factor was achieved by Rybin et al. by engineering of super cavity modes [2] in dielectric disk with subsequent experimental observation [3,4]. The underlying principle is an avoided crossing of two resonances of the dielectric disk under variation of the aspect ratio. The corresponding resonant modes have the same symmetry and interact through the continuum [5–9]. Along with the avoided crossing the hybridized resonant mode reveals multipolar conversion [10,11] that defines it as a super cavity mode. The resonant modes of different symmetry do not interact and therefore crossing of these resonances have no relevance for engineering of high-$Q$ resonances.

However, approaching of the second disk can remove the symmetry constraint for interaction of resonant modes of different symmetry. Because of interaction between the disks these resonant modes are hybridized forming bonding and anti-bonding resonant modes. The morphology of the anti-bonding resonant mode can become close to the Mie resonant mode of sphere with high orbital momentum at a certain distance between the disks. Respectively the $Q$ factor of the anti-bonding resonant mode can be greatly enhanced compared to the case of a single disk. Along with fundamental interest in the ways to enhance the $Q$ factor of the subwavelength dielectric resonators there is also a motivation caused by current experimental and technological facilities in fabrication of dielectric disks [11,12]. It is preferable to traverse over the distance between two disks that varies the effective height of disk dimer compared to the height of the isolated disk.

In contrast to the system of two parallel cylinders [13] or two microdisks [14] or two cuboids [15] the case of two coaxial disks preserves azimuthal index $m$ that allows to consider resonances specified by $m$ independently. In the present paper we focus on the case $m = 0$ in which the solutions are separated by polarization with $H_z = 0$ (TE modes) and $E_z = 0$ (TM modes) complemented by the case $m = 1$ for comparison. In the letter case the resonant modes leak into both E and H channels. That makes the case $m = 0$ more favorable with respect to the $Q$ factor. In what follows we consider two silicon disks with the permittivity $\epsilon = 12$ which have negligible material losses at the wavelength $\lambda = 1.5$ μm [16].

## 2. Avoided Crossing of Resonances of Single Dielectric Disk for Traversing Over Aspect Ratio

In general the resonant modes and their eigenfrequencies are given by solving the time-harmonic source-free Maxwell's equations [17,18]

$$\begin{pmatrix} 0 & -\frac{i}{\epsilon}\nabla\times \\ i\nabla\times & 0 \end{pmatrix} \begin{pmatrix} \mathbf{E}_n \\ \mathbf{H}_n \end{pmatrix} = k_n \begin{pmatrix} \mathbf{E}_n \\ \mathbf{H}_n \end{pmatrix} \tag{1}$$

where $\mathbf{E}_n$ and $\mathbf{H}_n$ are the EM field components defined in Ref. [18] as quasi normal modes which are also known as resonant states [19,20] or leaky modes [21]. Therefore the eigen-frequencies of resonances are complex $k_n = \omega_n + i\gamma_n$ with the quality factor given by ratio $Q_n = -\omega_n/2\gamma_n$. It is important that they can be normalized and the orthogonality relation can be fulfilled by the use of perfectly matched layers (PMLs) [18]. With the exception of a very restricted number of symmetrical particles (cylinders, spheres) Equation (1) can be solved only numerically, in particular, by COMSOL Multiphysics. Below we take the light velocity equal unit.

Because of the axial symmetry the solutions of the Maxwell Equation (1) are separated by the azimuthal index $m$. In what follows we consider the sector with $m = 0$ in which all components of EM field are independent of the azimuthal angle $\phi$. Then it follows from the Maxwell Equation (1) that for the TE polarization ($E_z = 0$) we have three nonzero components of EM field $E_\phi(r, z), H_r(r, z), H_z(r, z)$. Respectively for the TM polarization with $m = 0$ we have the nonzero components $H_\phi(r, z), E_r(r, z), E_z(r, z)$. For convenience of the reader, we first present the TE solutions for $E_\phi(r, z)$ and the TM solutions for $H_\phi(r, z)$ in Figure 1.

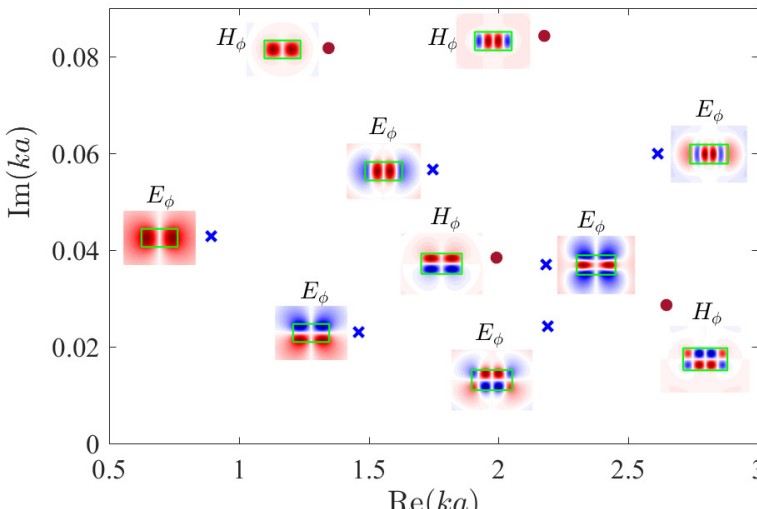

**Figure 1.** The resonant eigenfrequencies and corresponding resonant modes of dielectric disk with radius *a* equaled to its thickness *h* and permittivity $\epsilon = 12$ in silicon at $\lambda = 1.5$ μm. The TE/TM resonances are marked by crosses/closed circles. Thin green lines show disk.

Second, we demonstrate hybridization of the resonant modes symmetric relative to $z \rightarrow -z$ for avoided crossing of resonances TE solutions of Equation (1) for a single disk with the radius *a* and thickness *h* following Refs. [2,11]. The hybridization is shown in Figure 2a,b around the aspect ratio $a/h = 0.71$ . We also present the case of the avoided crossing of the anti-symmetric resonant modes in Figure 2c,d which results in the *Q* factor around 220 at the aspect ratio $a/h = 0.985$.

One can see that irrespective to the choice of the TE resonant modes in the isolated disk the pictures of avoided crossing are very similar. The interaction between two resonances through the radiation continuum results in a hybridization of the resonant modes.

The TM resonances demonstrate a different scenario for the avoided crossing.

One or another behavior of resonances for parametric traversing depends on difference between the couplings of the resonant modes with the radiation continuum, i.e., on the imaginary parts of the resonances. For small difference between couplings we observe the avoided crossing of resonances but for larger difference we observe the crossing of resonances [7,8]. However irrespective to the symmetry and polarization of the interfering resonant modes in the isolated disk one can see from the insets in Figures 2 and 3 that the hybridized modes with maximal *Q* factor have a morphology close to the morphology of the Mie resonant mode in sphere. These modes are highlighted by yellow open circles and have the orbital momentum $l = 3$ in Figure 2a, $l = 4$ in Figure 2c, and $l = 3$ in Figure 3a. Similar effects were observed recently by Huang et al. for long rod of rectangular cross-section for its limiting to a square [22]. Correspondingly one can observe multipolar conversion for these modes [11,12]. In what follows we show that similar morphology of the Mie resonant modes with considerably higher *l* can be reached for two disks to result in unprecedent enhancement of the *Q* factor.

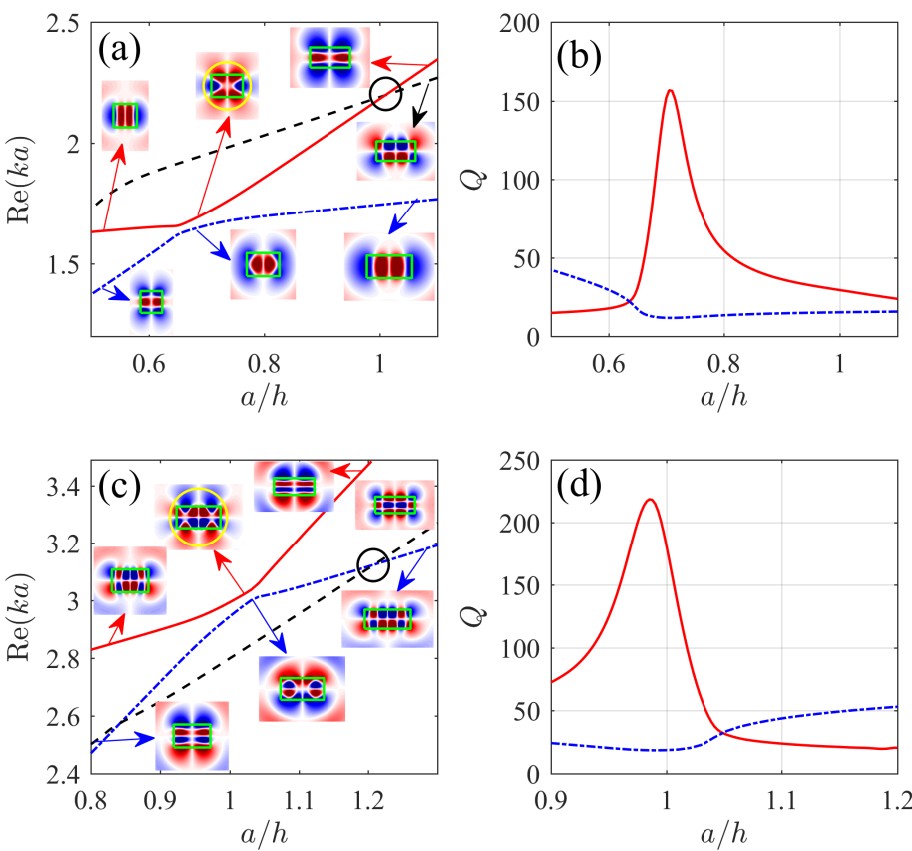

**Figure 2.** (**a**) Avoided crossing of two TE resonances (red and blue solid lines) whose modes are symmetric relative to $z \to -z$ versus the aspect ratio $a/h$ in isolated silicon disk. The anti-symmetric resonant mode shown black dash line is not involved into avoided crossing. Insets show evolution of hybridized mode (the tangential component of electric field $E_\phi$). (**b**) The behavior of the $Q$ factor of the corresponding resonances vs aspect ratio. (**c**) Avoided crossing of two anti-symmetric (red and blue dash lines) TE resonances. The symmetric resonant mode shown black solid line does not participate in avoided crossing. (**d**) The $Q$ factors vs aspect ratio.

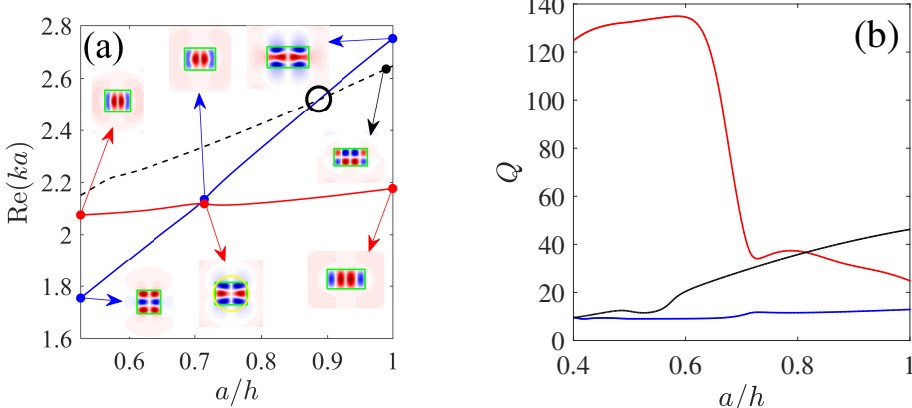

**Figure 3.** (**a**) Avoided crossing of the TM-resonances in isolated disk. Insets show the profiles of the component of magnetic field $H_\phi$. (**b**) The behavior of the $Q$ factors.

## 3. Two-Fold Avoided Crossing

In the system of two dielectric disks we have a second parameter to vary, the distance between the disks, *L*. This distance is measured between volumetric centers of disks as shown in Figure 4, therefore minimal distance is *h*. First of all

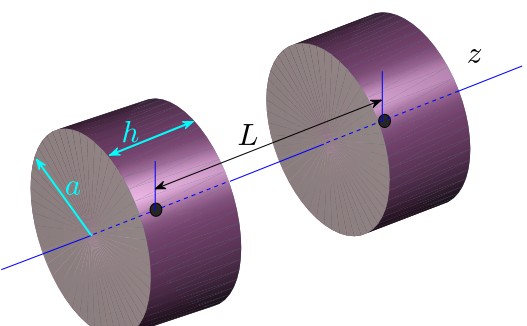

**Figure 4.** Two coaxial dielectric disks with $\epsilon = 12$ separated by distance *L* measured between the centers of disks.

variation of the distance provides technological advance over variation of the aspect ratio of isolated disk because of continual variation of the effective height of disk's dimer equal $h + L$. However, the presence of the second disk brings on two fundamental aspects. The first is the interaction between the disks. The radiation of the first disk is scattered by the second disk resulting in coupling between the disks that lifts the degeneracy of the resonances of two disks with further avoided crossing of resonances for variation of *L* [23–26]. The second aspect related to avoided crossing of orthogonal resonant modes highlighted by black circles in Figure 2 is novel. For isolated disk these events have no effect because of their opposite symmetry relative to inversion of disk's axis $z \to -z$. However as soon as the second disk is approaching this symmetry prohibition is removed.

At first, we show in Figure 5 the behavior of resonances with distance between disks which are optimized for maximal *Q*-factor at the aspect ratio $a/h = 0.71$.

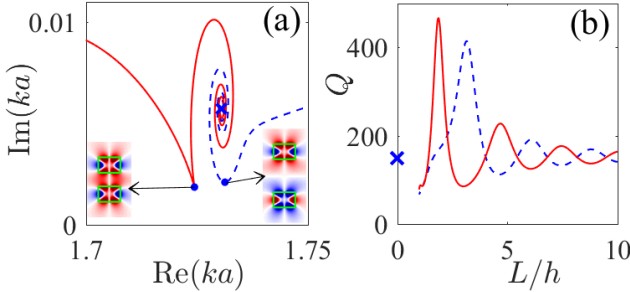

**Figure 5.** (**a**) Evolution of the hybridized bonding (solid) and anti-bonding (dash) resonances under variation of the distance between the disks *L* at the aspect ratio $a/h = 0.71$. Insets show the tangential component $E_\phi$ of electric field. Cross marks the resonance of the isolated disk. (**b**) Evolution of the *Q* factors vs the distance.

At $L \gg 2h$ the resonances are almost degenerate and marked by cross in Figure 5a. When the disks become closer they are coupled because of radiation of leaky resonant modes by one disk and consequent scattering by the another. The scattering processes give rise to spiral evolution of complex eigenfrequencies as shown in Figure 5a because of the coupling $e^{ikL}/L^2$ which hybridizes the resonant modes of separate disks as the bonding and anti-bonding leaky resonant modes [23,25,27,28]. For large distances between disks ($L \gg 2h$) and provided that neighboring resonances of the disk are not overlapped the

bonding and anti-bonding modes can be presented as the symmetric and anti-symmetric superpositions of the resonant modes of individual disks $\psi_n$:

$$\Psi_{n;s,a}(\vec{r}) \approx \psi_n(\vec{r}_\perp, z - \frac{1}{2}L\vec{z}) \pm \psi_n(\vec{r}_\perp, z + \frac{1}{2}L\vec{z}) \qquad (2)$$

where $\vec{z}$ is the unit vector along the z-axis, and $\psi_n$ are resonant modes of disk shown in Figure 1. These modes are illustrated in the insets of Figure 5a. That scenario for engineering of the Q-factor based on avoided crossing of the resonant modes of the same symmetry was considered in Ref. [25]. For the present case of silicon disks we gain the Q factor amounted to 450.

A different scenario based on avoided crossing of orthogonal resonant modes highlighted by black circles in Figure 2 is novel. For isolated disk these modes can cross but that has no effect because of their opposite symmetry relative to inversion of disk's axis $z \to -z$. However for approaching of second disk this symmetry restriction is removed. Evolution of the four selected resonances with distance $L$ between disks is shown in Figure 6.

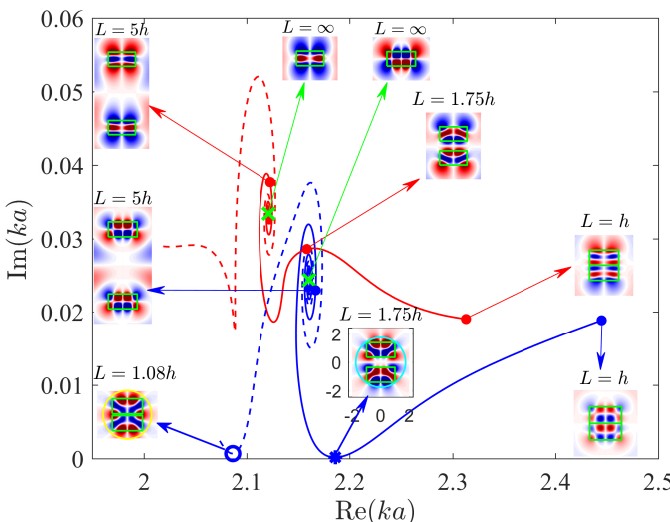

**Figure 6.** Avoided crossing of resonances originated from coupling of orthogonal resonances of isolated disk shown in Figure 2a for variation of distance between disks for $a = 0.5$ μm, $a/h = 0.96$. Solid/dash lines show the anti-bonding/bonding resonances. Insets show the profiles of tangential component of electric field $E_\phi$.

When the distance is large enough the resonances marked by green crosses are degenerate. We define the corresponding modes as $\psi_1(\vec{r})$ and $\psi_2(\vec{r})$ shown in Figure 6 in the upper insets at $L = \infty$. With approaching of the disks both resonant modes are hybridizing as given by Equation (2) forming the bonding and anti-bonding configurations $\psi_{n,s,a}$. In order to not encumber figure we show only the anti-bonding resonant modes which result in maximal Q-factors. The insets of Figure 6 illustrate examples of the anti-bonding modes at $L/h = 5$. However, with further approaching of the disks the approximation (2) ceases to be correct because of overlapping of the resonances $\psi_1$ and $\psi_2$. One can observe noticeable distortions from the superpositions (2) at $L/h = 1.75$ inside the disks. At this distance and the aspect ratio $a/h = 0.96$ the anti-bonding mode highlighted by blue circle in Figure 6 has extremely low resonant width. Respectively, one can observe extremely high peak of the Q factor around 5500 in Figure 7 at the vicinity of the point $L/h = 1.68$ and $a/h = 0.96$.

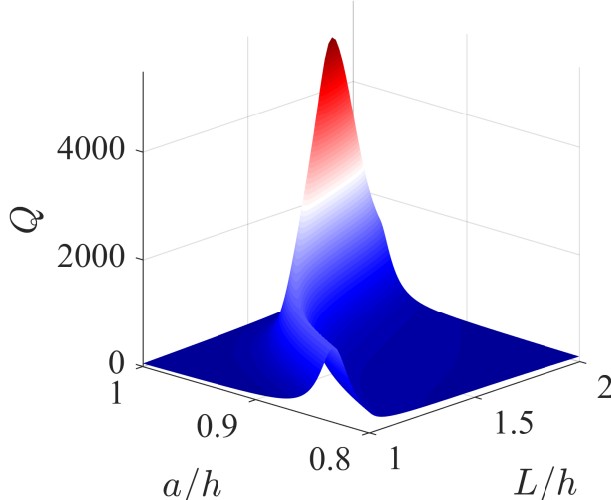

**Figure 7.** The $Q$ factor vs aspect ratio and distance between disks of the anti-bonding resonant mode shown by blue line in Figure 6.

The reason of extremely small radiation losses of the anti-bonding mode at the point marked by star in Figure 6 is related to a morphology of the mode. One can see it is close to the morphology of the Mie resonant mode with orbital momentum $l = 6$ of an effective sphere with volume $\pi(h + L)a^2$. That sphere is highlighted in the inset in Figure 6 by blue open circle. Although inside the effective sphere the morphology of the anti-bonding mode can differ from the Mie resonant mode of a true sphere it is important that they have a very similar morphology on the surface of the sphere. As one can see from Figure 6 there is another point with minimal radiation losses marked by yellow open circle at $L = 1.08h$. We see that the resonant mode has a morphology of the Mie resonant mode with the lower orbital momentum $l = 5$. Respectively the radiation losses exceed the former case with $l = 6$ at $L = 1.75h$.

These events of extremal enhancement of the $Q$ factor because of formation of the Mie like resonant modes in the system of two coaxial disks are not unique. For example, Figure 2c shows another case of crossing of the resonances of opposite symmetry at the aspect ratio $a/h = 1.17$ of the isolated disk. Figure 8a demonstrates an avoided crossing of resonances for variation of $L$. Similar to the case shown in Figure 6 the bonding and anti-bonding resonant modes both turn to the Mie-like modes with $l = 7$ and $l = 8$ highlighted by blue and yellow circles at respective distances shown above the insets. At these distances the resonant modes reach extremely large $Q$ factors, 3000 and 15,000. Figure 8b shows as the $Q$-factor of the resonant mode shown by brown dash-dotted line in Figure 8a is optimizing for two-fold variation of aspect ratio and distance.

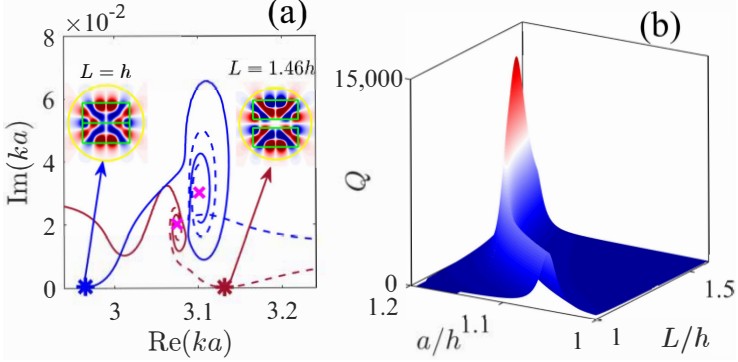

**Figure 8.** (**a**) Evolution of the higher lying TE resonances in traversing with the distance between the disks for $a/h = 1.17$. (**b**) The Q factor vs the distance between the disks $L/h$ and their aspect ratio $a/h$.

In addition we demonstrate in Figure 9 engineering of the TM Mie-like resonant modes by hybridization of the TM modes $H_\phi$ that also enormously enhances the $Q$ factor.

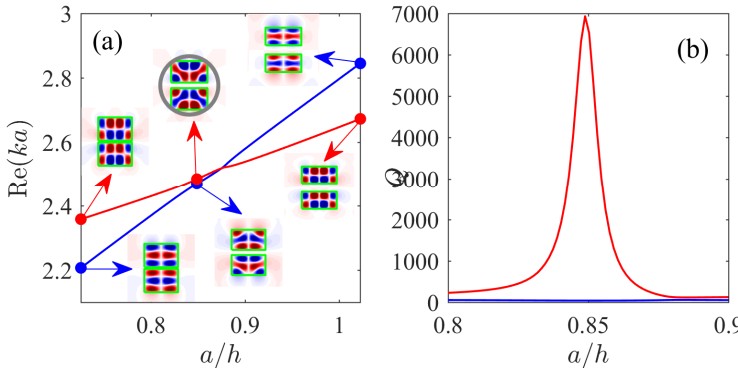

**Figure 9.** Evolution of the TM resonances (**a**) and $Q$ factor (**b**) in traversing with the height of the disks at $L/a = 1.48$. Insets shows profiles of the TM resonant modes $H_\phi$.

Thus, the recipe of extremely high $Q$ factor in the system of two coaxial disks is the following. First, find crossing of two orthogonal resonant modes in the isolated disk. Second, under variation of the distance between disks achieve the anti-bonding resonant mode with morphology close to the Mie-like resonant modes in the effective sphere. In Table 1 we collected some Mie-like hybridized resonant modes and their $Q$ factors.

**Table 1.** Parameters of the high-$Q$ resonant modes of two disks with radius $a = 0.5$ μm and their shapes of tangential components $E_\phi$ for TE modes or $H_\phi$ for TM modes. $D = [4a^2 + (h+L)^2]^{1/2}$.

| Polarization | Re($ka$) | $\lambda$ [μm] | $a/h$ | $L/a$ | $\lambda/D$ | $Q$ | Mode Shapes |
|:---:|:---:|:---:|:---:|:---:|:---:|:---:|:---:|
| TE | 1.724 | 1.822 | 0.71 | 1.873 | 0.948 | 450 | |
| TE | 1.7314 | 1.814 | 0.71 | 3.15 | 0.729 | 420 | |
| TE | 2.0865 | 1.506 | 0.96 | 1.08 | 1.033 | 1450 | |
| TE | 2.186 | 1.437 | 0.96 | 1.75 | 0.837 | 5634 | |
| TE | 3.13 | 1.004 | 1.17 | 1.46 | 0.656 | 15656 | |
| TM | 2.485 | 1.264 | 0.85 | 1.48 | 0.76 | 6930 | |

## 4. Cancellation of the Lowest-Order Terms in a Multipole Radiation for Avoided Crossing

The sphere has the minimal surface at given volume and therefore minimal radiation losses. The Mie resonant modes in the sphere given by spherical harmonics are classified by the orbital momentum $l$ and have a tendency of exponential decreasing of the resonant width with $l$ as $e^{-\alpha/\sqrt{l}}$ where $\alpha$ some constant defined in [29]. That explains extremely large $Q$ factors of the anti-bonding resonant modes whose morphologies are similar to the Mie-like modes with high orbital index. In this section we show full agreement with the multipole analysis of the radiated power similar to the case of the isolated disk considered in Refs. [11,12].

Any solution of the Maxwell equations can be expanded in terms of electric and magnetic spherical harmonics [30]

$$\mathbf{E}(\mathbf{x}) = \sum_{l=1}^{\infty} \sum_{m=-l}^{l} [a_{lm}\mathbf{M}_{lm} + b_{lm}\mathbf{N}_{lm}]. \tag{3}$$

Then the relative radiated power of each electric and magnetic multipole of order $l$ is given by squared amplitudes of expansion [30]

$$P_{lm} = P_{lm}^{TE} + P_{lm}^{TM} = P_0^{-1}[|a_{lm}|^2 + |b_{lm}|^2] \tag{4}$$

where $P_0$ is the total power radiating through the sphere with large radius

$$P_0 = \sum_{l=1}^{\infty} \sum_{m=-l}^{l} [|a_{lm}|^2 + |b_{lm}|^2]. \tag{5}$$

It is intuitively clear that the sharp enhancement of the $Q$ factor is a result of cancellation of the lowest-order terms in the multipole expansion of the far-field radiation, distinct from the near-field multipole symmetry [11,31].

For the case of avoided crossing with variation of the aspect ratio of the disk Chen et al. has interpreted that significant Q-factor enhancement is the result of conversion of electric dipole radiation into octuple one as shown in Figure 10. The case corresponds to the avoided crossing of resonant modes shown in Figure 2a.

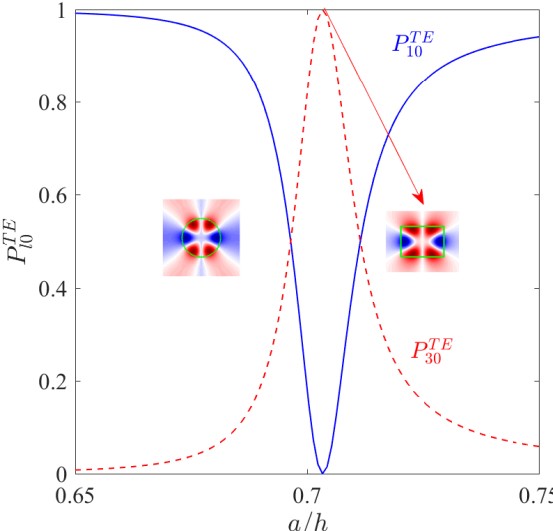

**Figure 10.** Evolution of the TE multipole radiated powers from one disk vs the aspect ratio in isolated disk for the case shown in Figure 2.

The insets in Figure 10 clearly demonstrate that the field configuration at the maximal conversion is very close to the Mie octuple resonant mode in the effective dielectric sphere with the radius given by the equality of volumes of the sphere and disk at $a/h = 0.706$: $4\pi R^3/3 = \pi a^2 h$. Figure 11 demonstrates that in spite of different behavior of the TM resonances shown in Figure 5 the radiated multipole powers also undergo similar conversion as shown in Figure 10. The corresponding Mie octuple TM resonant mode of the sphere with frequency $kR = 1.94$ and $Q = 194$ presented by the component $H_\phi$ is similar to the Mie engineered TM hybridized mode of the disk with frequency $ka = 2.12$ and $Q = 135$ as shown in Figure 11.

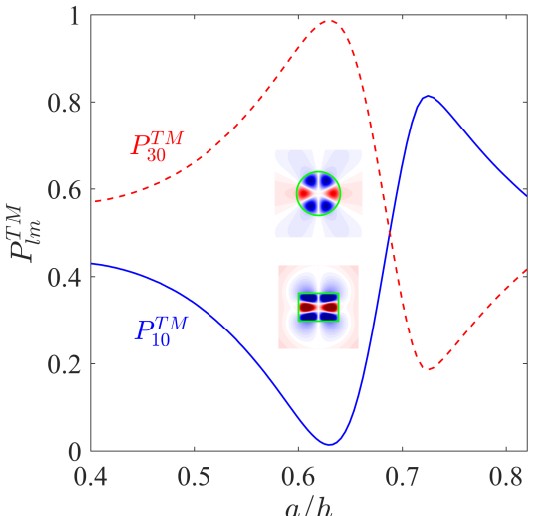

**Figure 11.** Evolution of the multipole TM powers from disk vs the aspect ratio for the case shown in Figure 7.

Next, let us consider the case of two disks. We start with the case presented in Figure 6 when for avoided crossing of the TE resonances of opposite symmetry the $Q$ factor approaches 5500. Corresponding evolution of the multipole powers $P_{l0}^{TE}$ is presented in Figure 12.

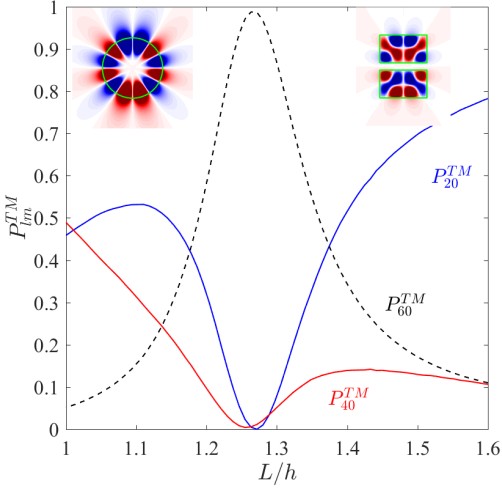

**Figure 12.** Evolution of the radiated power from two disks vs the distance between disks for the case shown in Figure 6 ($h = 1.038a$).

Similar to the isolated disk we introduce equivalent sphere by equality of the volume of disk's dimer and sphere $4\pi R^3/3 = \pi a^2(h + L)$. The $Q$ factor 5500 for the hybridized resonance $ka = 2.2$ is substantially smaller than the $Q$ factor 23,100 of the true Mie resonant mode $l = 6, m = 0$ with the frequency $kR = 2.68$ of the equivalent sphere with the radius $R = 0.83\mu$ shown in the left inset of Figure 12. In spite of that one see that the hybridized anti-bonding resonant mode of two disks shown in the right inset of Figure 12 has the same morphology that explains so high $Q$ factor of two disks. While for the avoided crossing of TE resonances of the same symmetry shown in Figure 5 the hybridized bonding resonant modes have the morphology cardinally different from the Mie resonant modes of the sphere. As a result the bonding resonances have no extremely high $Q$ factors.

It is evident from Figures 10–12 that the sharp $Q$ factor enhancement is intrinsically connected with multipole conversions from lower to higher orders. Destructive interference between two resonances underlie these phenomena when the resonances undergo avoided crossing. For that process the system supports the hybridized modes which become maximally close to the Mie resonant modes of the sphere with the high orbital momentum indices.

## 5. Sector $m = 1$

Above engineering of the Mie-like resonances of two coaxial disks was considered in the sector $m = 0$ in which the resonant modes radiate only into the TE or TM continua. In the sectors with nonzero $m$ the polarizations are not separated and therefore one could expect that the $Q$ factors of the resonant modes would be less than the sector $m = 0$. Figure 13a,b shows the evolution of resonances with the aspect ratio $a/h$ and respectively the behavior of the $Q$ factor in the sector $m = 1$ at the distance $L = 1.3a$ tuned to the maximal quality factor. Comparison to the case $m = 0$ in Figure 2 indeed shows that radiation into both continua substantially increases radiation losses of the two disks as compared to the case of $m = 0$ shown in Figure 7. Figure 13c with plots of the multipole powers reveals that avoided crossing with traversing over two parameters is not enough to cancel simultaneously electric quadruple and magnetic dipole radiated powers.

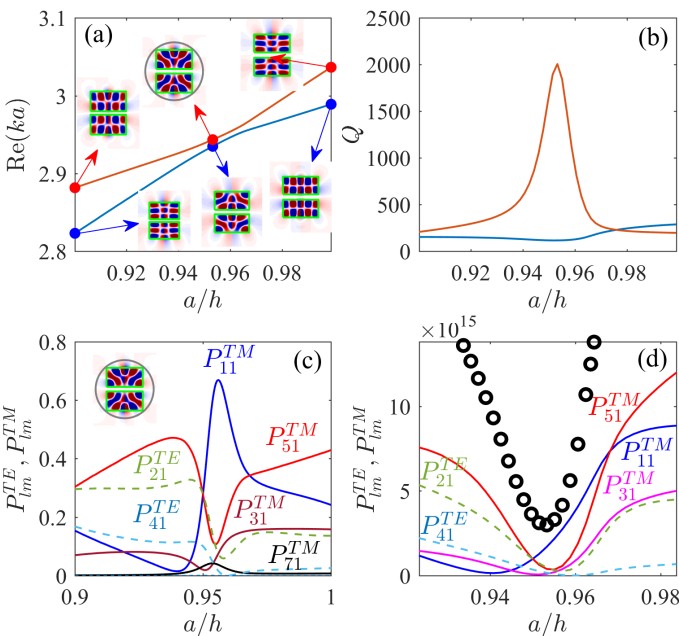

**Figure 13.** Avoided crossing of two resonances (**a**) and the $Q$ factor (**b**) versus the aspect ratio $a/h$ at $L = 1.3a$ in the sector $m = 1$. Evolution of the radiated multipole powers normalized by (**c**) total power and (**d**) by energy inside the disks.

If to normalize the radiated power through the total energy accumulated inside the disks we observe the resonant dip in the power as shown in Figure 13d by open circles.

## 6. Conclusions and Outlook

When two resonances of the open resonator are traversed over some parameter they undergo an avoided crossing (anti crossing) with hybridization of the resonant modes. Typically the real and imaginary parts of resonances, both, undergo repulsion up to that one of the hybridized modes acquires the imaginary part significantly less compared to the case far from the region of the avoided crossing. That property of the avoided crossing was numerously used to achieve high $Q$ factor by variation of the aspect ratio of the isolated dielectric resonator [2–4] or two resonators by changing of the distance between them [9,13,14,32]. In the present paper we studied avoided crossing of resonances and respectively the $Q$ factor traversing over two parameters, the aspect ratio of disks and distance between them. One could expect that the maximal enhancement of $Q$ factor will be multiplication of gains achieved by changing over each parameter independently. Indeed, if, first, to vary the aspect ratio with the aim to achieve maximal $Q = 158$ at $a/h = 0.71$ as it was reported by Rybin et al [2]. Then variation of the distance between disks results in successive gain in the $Q$ factor as shown in Figure 5 and given in Table 1. This result was reported in our previous paper [25] and gives therefore triple gain compared to the case of the isolated disk. However as seen from the first row in Table 1 the total height $h + L$ of the system also exceeds the height of the disk three times.

In the present manuscript we consider different scenario based on crossing of the resonant modes which were orthogonal in the isolated disk and therefore could not contribute into avoided crossing for variation of the aspect ratio. Events of crossing of these modes are highlighted by black circles in Figures 2 and 3. One can see that these points are rather far from former point $a/h = 0.71$. The presence of a second disk removes the symmetrical constrain and gives rise to a new series of avoided crossings shown in Figures 6 and 7. What is important only these avoided crossings result in anti-bonding resonant modes which have morphology close to the Mie-like resonant modes with high orbital indices of effective sphere as Figures 6, 8, and 10 and Table 1 demonstrate. That explains extremely high $Q$ factor of these anti-bonding resonant modes. The resulting $Q$ factor exceeds the $Q$ factor of the isolated disk by two or three orders of magnitude. The avoided crossing of the TM resonances gives similar results as in Figure 10 and the last row in Table 1.

There is a useful tool to understand of the nature of the extremely high quality factor for the avoided crossing through multipole expansions [30]. That tool shed light on the origin of the high $Q$ factor in the isolated disk [3,11] and the origin of bound states in the continuum [33]. In the present case of two disks we also observe that extremal Q-factor enhancement is attributed to strong redistribution of radiation that originates from multipole conversions from lower to higher orders for the case of zero azimuthal index $m = 0$. However, for $m = 1$ disks radiate in both continua, TE and TM. As a result such a redistribution turns out to be not sufficient to strongly suppress radiation and enhancement of the $Q$ factor is not so impressive.

Note that these results refer to low lying subwavelength resonances which are important for numerous applications. Thus, such a tuned dimer successfully meets the request of high sensitivity and selectivity sensors which can be integrated in microsystems [34]. Since the unprecedent enhancement of the $Q$ factor is the result of lifting of symmetry restrictions in the system of two coaxial disks we expect similar phenomenon for the disks of other materials with different permittivity with simple scaling of the $Q$ factor in the form $Q \sim \epsilon^\alpha$ where for particular case of isolated disk $\alpha$ was estimated as $\alpha = 3.2$ according [2].

It is clear that the phenomenon of the avoided crossing and respective enhancement of the $Q$ factor would occur with particles of arbitrary shape when the distance between them is varied. The case of two coaxial disks simplifies computations because the solutions with different azimuthal index $m$ are independent. Despite that the introduction of a second disk extends the size of the dielectric resonator more than two times, it provides an easier

technological way to achieve avoided crossing. That strategy resulted in unprecedent enhancement of the $Q$ factor by two and three orders of magnitude compared to the case of an isolated disk with a given aspect ratio.

**Author Contributions:** Investigation, E.B. and A.S.; data curation, K.P. All authors have read and agreed to the published version of the manuscript.

**Funding:** This work was supported by RFBR grant 19-02-00055.

**Institutional Review Board Statement:** Not applicable.

**Informed Consent Statement:** Not applicable.

**Acknowledgments:** The author thanks D.N. Maksimov, Vladimir Tuz and Yi Xu for discussions.

**Conflicts of Interest:** The authors declare no conflict of interest.

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
