# Peer review of "Mie Resonance Engineering in Two Disks"

_photonics, doi:10.3390/photonics8020049_

Round 1

Reviewer 1 Report

The manuscript presents a very comprehensive and detailed study of the resonances in dimer of dielectric coaxial wavelength-scale disks. The Authors consider different hybridization scenarios between modes localized in each disk and provide general guidelines for harnessing exceptionally high-Q modes in this system.

With no doubts, the manuscript is of interest for the photonics community. I recommend it for publication in Photonics.

While the scientific content of the manuscript is coherent and clearly presented, there is a very large number of typos and issues. I strongly encourage the Authors to revise the English and the text overall, only on the first page I've spotted at least 4 issues:

l.2: has -> have
l.19: "with" is redundant
l.32: fir -> for; ots -> its
l.36: repeated "these"

Author Response

Reviewer 1:
The manuscript presents a very comprehensive and detailed study of the resonances in dimer of dielectric coaxial wavelength-scale disks. The Authors consider different hybridization scenarios between modes localized in each disk and provide general guidelines for harnessing exceptionally high-Q modes in this system.
With no doubts, the manuscript is of interest for the photonics community. I recommend it for publication in Photonics.
-------------------------------------------------------------------------------------------------------------------------------------------
We are pleased by high evaluation of our manuscript and thank the Referee.

While the scientific content of the manuscript is coherent and clearly presented, there is a very large number of typos and issues. I strongly encourage the
Authors to revise the English and the text overall, only on the first page I've spotted at least 4 issues:

l.2: has -> have
l.19: "with" is redundant
l.32: fir -> for; ots -> its
l.36: repeated "these"
------------------------------------------------------------
We have corrected typos.

Reviewer 2 Report

In this paper, the authors studied how high-Q resonances can be formed for a pair of coaxial silicon disks. It is a continuation of their previous work Ref. [24]. They showed how avoided crossing of "orthogonal resonant modes" (of a single disk) leads to new anti-bonding resonant modes with very high Q values. They also discussed how the high-Q modes have similarity with high-Q resonant modes of a sphere (Mie resonances). They results are new, interesting, and have potential useful applications. The presentation can be improved. There are some minor problems with the use of English. Since the main interest is in high-Q resonances of subwavelength structures, the authors may provide a table, listing the resonances with a local max of Q. Such a table may include: h/a, L/a, D/Wavelength, Q, where D = sqrt( 4a^2 + (h+L)^2), is the max size of the dimer, wavelength is the resonance wavelength, related to the resonant frequency.

Author Response

The Reviewer 2:
In this paper, the authors studied how high-Q resonances can be formed for a pair of coaxial silicon disks. It is a continuation of their previous work Ref. [24].
They showed how avoided crossing of "orthogonal resonant modes" (of a single disk) leads to new anti-bonding resonant modes with very high Q values.
They also discussed how the high-Q modes have similarity with high-Q resonant modes of a sphere (Mie resonances). They results are new, interesting,
and have potential useful applications.
---------------------------------------------------------------
We thank the Referee for positive evaluation of our manuscript.

The presentation can be improved. There are some minor problems with the use of English. Since the main interest is
in high-Q resonances of subwavelength structures, the authors may provide a table, listing the resonances with a local max of Q. Such a table may include:
h/a, L/a, D/Wavelength, Q, where D = sqrt( 4a^2 + (h+L)^2), is the max size of the dimer, wavelength is the resonance wavelength, related to the resonant frequency.
--------------------------------------------------
We are very grateful for these suggestions especially for the idea of table which will, indeed, useful for experimentalists. We have added the table.

Reviewer 3 Report

In the given manuscript, Bulgakov et. al. investigated avoided crossing of the resonant modes of two coaxial disks. The authors first present resonant modes of the single isolated disk with field profile and corresponding Q factor, and further, avoided crossing of the resonances for the two disks with a various aspect ratio of the disks and distance between disks. The authors tried to suggest abundant images of the field profile to visualize the various resonant modes and their distortion and Q factor to quantify the effect of the resonance. Unfortunately, however, the submitted manuscript with the current form is insufficient for publication in Photonics because overall the manuscript has a readability problem at the sentence level and the paragraph level. (Many of the sentences are long and lengthy. Conclusion and Outlook section is lengthy.) And the sentences cannot describe or explain the figures. As a result, it is difficult to catch the message of the contents. Also, the authors show many resonances with various situations but they just seem to be listed, not arranged to support any conclusion. It is clearly worthwhile to investigate the avoided crossing of the resonances, I would like to recommend to trim the sentences.

Author Response

The Reviewer 3:
In the given manuscript, Bulgakov et. al. investigated avoided crossing of the resonant modes of two coaxial disks. The authors first present resonant modes of the
single isolated disk with field profile and corresponding Q factor, and further, avoided crossing of the resonances for the two disks with a various aspect ratio
of the disks and distance between disks. The authors tried to suggest abundant images of the field profile to visualize the various resonant modes and their
distortion and Q factor to quantify the effect of the resonance. Unfortunately, however, the submitted manuscript with the current form is insufficient for publication
in Photonics because overall the manuscript has a readability problem at the sentence level and the paragraph level. (Many of the sentences are long and lengthy.
Conclusion and Outlook section is lengthy.) And the sentences cannot describe or explain the figures. As a result, it is difficult to catch the message of the contents.
Also, the authors show many resonances with various situations but they just seem to be listed, not arranged to support any conclusion. It is clearly worthwhile
to investigate the avoided crossing of the resonances, I would like to recommend to trim the sentences.

----------------------------------------------------------------------------------------------------------
We thank the Referee for careful reading of manuscript and criticism. Once more we corrected the text and tried to trim and shorten the sentences.

Round 2

Reviewer 3 Report

I'd like to appreciate the author's effort in the revised version. I had thought several errors are minor and could be corrected when the authors trimmed the sentences. However, many imperfections still remain and need to correct.
1. In line 79 on p2, there are a and h for the first time but there is no definition of them. Their definitions are introduced later on, in Fig. 4 on p4. 
2. In line 81 on p2, does the resonant mode for a/h=1.02 indicate a maximum point of the red dotted line in Fig. 2(d)? The maximum point seems not to be located at a/h=1.02 and Q factor does not show 220 at a/h=1.02.
3. Section 2 explains the behavior of a single dielectric disk, prior to two-fold avoided crossing for two coaxial dielectric disks in Section 3. However, the explanation and figure caption of Fig. 2 is not kind and legible. 
1) Is 'a' in Re(ka) in Fig. 1 same as 'a' in aspect ration 'a/h'? It makes readers confused. 
2) There is no explanation of red, blue, and black lines in Fig. 2 and Fig. 3 in the main text or figure caption. 
3) What is related to l=4 (in line 92 on p3) and l=8 (in line 93 on p4) in Fig. 2 and Fig. 3?
4) In figure caption for Fig. 1, '(closed circles)' can be removed. I think both red closed circles and blue crosses indicate resonant eigenfrequencies. In figure caption for Fig. 2(a), 'their Q factors' can be removed. 
5) What do white circles mean in insets in Fig. 2(a) and Fig. 2(c)?
4. In line 104 on p4, the authors mentioned that there are two fundamental aspects the second disk induces. The first one is an interaction between the disks as the just next sentence said. I think another aspect, the second one would be better to be introduced in the following part within the same paragraph for easy understanding.
5. What is 'apporoximation (2)' in line 125 on p6? What is 'superposition (2)' in line 127 on p6?
6. In figure caption for Fig. 6, '(a)' can be removed. Is it meaningful? There is no explanation for green circle in inset (L=1.75h) of Fig. 6.
7. In figure caption for Fig. 7, ')' can be removed. Unlike figure caption for Fig. 7 ('resonant mode highlighted by white circle in inset of Fig. 6'), there is no white circle in inset of Fig. 6. In Fig. 7, '(e)' can be removed. 
8. In line 134 on p6, l is orbital index while in line 141 on p6, l is orbital momentum. To avoid confusion, it would be better to use only one term. 
9. Fig. 6(a) mentioned in line 147 on p6 does not exist. 
10. It is needed to check the figure number, 'Fig. 10', in line 150 on p6. Isn't the sentence about Fig. 9?
11. Fig. 10 and Fig. 11 are same. I think Fig. 10 in the previous manuscript is missed. 
12. After line 178 on p9, parenthesis ')' is missed. 
13. In line 182 on p10, what is Fig. 6(a) and (b)? I think the figure number is incorrect. 
14. The author mentioned an extremely large Q factor of 23100 in line 187 on p10. Can the reader find the value in the manuscript? 
15. In line 189 on p10: dik--> disk

Author Response

We would like to deeply thank the Referee 3 for careful and scrupulous reading of our manuscript and his/her comments.
